# The Absolute Monocyte Count at Diagnosis Affects Prognosis in Myelodysplastic Syndromes Independently of the IPSS-R Risk Score

**DOI:** 10.3390/cancers15143572

**Published:** 2023-07-11

**Authors:** Tobias Silzle, Sabine Blum, Annika Kasprzak, Kathrin Nachtkamp, Martina Rudelius, Barbara Hildebrandt, Katharina S. Götze, Norbert Gattermann, Michael Lauseker, Ulrich Germing

**Affiliations:** 1Department of Medical Oncology and Hematology, Cantonal Hospital St. Gallen, 9007 St. Gallen, Switzerland; 2Service and Central Laboratory of Hematology, University Hospital of Lausanne and Lausanne University, 1011 Lausanne, Switzerland; 3Department of Hematology, Oncology, and Clinical Immunology, Heinrich Heine University Düsseldorf, 40225 Düsseldorf, Germany; 4Institute of Pathology, Faculty of Medicine, LMU Munich, 80337 Munich, Germany; 5Department of Human Genetics, Heinrich Heine University Düsseldorf, 40225 Düsseldorf, Germany; 6Department of Internal Medicine III, Klinikum Rechts der Isar, Technical University of Munich, 81675 Munich, Germany; 7Institute for Medical Information Processing, Biometry and Epidemiology, Faculty of Medicine, LMU Munich, 81377 Munich, Germany

**Keywords:** myelodysplastic syndrome, prognostication, absolute monocyte count, revised international prognostic scoring system

## Abstract

**Simple Summary:**

The revised international prognostic scoring system (IPSS-R) represents the standard tool for prognostication in myelodysplastic syndromes (MDSs). It considers the degree of cytopenias together with the bone marrow blast count and the results of metaphase cytogenetics. Monocytes are a subgroup of white blood cells involved in host defense and tissue repair or remodelling. The goal of our study was to assess the prognostic impact of the absolute monocyte count (AMC) at the time point of diagnosis in patients with MDS. We found an IPSS-R-independent prognostic impact of the AMC, both when assessed as a continuous variable and when MDS patients with a low (<0.2 × 10^9^/L) or a higher (>0.4 × 10^9^) AMC were compared to MDS patients with an AMC of 0.2–0.4 × 10^9^/L. A low AMC was associated with a higher risk of transformation to acute myeloid leukemia. Hence, considering the AMC might help to identify MDS patients who could benefit from more intense treatment strategies.

**Abstract:**

The absolute monocyte count (AMC) is associated with mortality in a variety of medical conditions. Its prognostic impact in myelodysplastic syndromes (MDSs) is less well studied. Therefore, we investigated its potential prognostic value in a cohort from the Düsseldorf MDS registry in relationship to the revised international prognostic scoring system (IPSS-R). An AMC below the population’s median (<0.2 × 10^9^/L) was associated with several adverse disease features such as lower haemoglobin levels, lower count of neutrophils and platelets, and a higher percentage of blasts in the bone marrow. MDS patients with an AMC < 0.2 × 10^9^/L had a significantly higher risk of progression into acute myeloid leukemia (AML). In a univariate, proportional hazards model the effect of the AMC as a continuous variable was modelled via p-splines. We found a U-shaped effect with the lowest hazard around 0.3 × 10^9^/L. Accordingly, an AMC within the last quartile of the population (0.4 × 10^9^/L) was associated with a reduced overall survival independently of IPSS-R, but not with the risk of secondary AML. Considering monocytopenia as a risk factor for AML progression in MDS may provide an additional argument for allogeneic transplantation or the use of hypomethylating agents in patients who are not clear candidates for those treatments according to current prognostic scoring systems and/or recommendations. Further studies are needed to assess the prognostic impact of the AMC in the context of prognostic scoring systems, considering the molecular risk profile, and to identify the mechanisms responsible for the higher mortality in MDS patients with a subtle monocytosis.

## 1. Introduction

The myelodysplastic syndromes (MDSs) are a heterogeneous group of clonal myeloid haemopathies, primarily observed in the elderly population. Hallmarks of these diseases include cytopenias due to an ineffective haematopoiesis and a variable risk of leukemic transformation due to genetic and epigenetic changes in haematopoietic stem cells and alterations within the haematopoietic niche [1,2]. Their classification is based on morphology (number of dysplastic cell lines, number of blasts, presence or absence of ringed sideroblasts) and the presence of certain cytogenetic or molecular aberrations [3,4,5].

Risk stratification with regard to the probability of leukemic transformation and death related to bone marrow failure is essential for newly diagnosed MDS patients [6]. The degree of cytopenias, the bone marrow blast count, and the presence and type of cytogenetic changes determined by conventional metaphase cytogenetics are factors taken into account by the standard tools used for prognostication, the international prognostic scoring system (IPSS) [7] and its revised version (IPSS-R) [8]. Recently, the addition of the molecular risk profile, as determined by next generation sequencing (NGS), has been shown to allow a refined prognostication in addition to the conventional factors. The IPSS-molecular takes into account mutations of 31 different genes, with *TP53*^multihit^, *MLL*^PTD^, and *FLT3* mutations as top genetic predictors of adverse outcomes, and mutations in genes such as *ASXL1*, *BCOR*, *EZH2*, *NRAS*, *RUNX1*, *STAG2*, and *U2AF1* being significantly associated with an adverse risk [9].

Patients with higher-risk MDSs are candidates for allogeneic haematopoietic stem cell transplantation (HSCT) since the risk of leukemic transformation is high and overall survival (OS) is short even without transformation. Best supportive care and improvement of cytopenias are the cornerstones of therapy for lower-risk MDS patients [1].

Monocytes are a functionally heterogeneous group of mononuclear phagocytic cells that contribute in many ways to host defense and tissue repair or remodelling [10]. A low absolute monocyte count (AMC) has been shown to be of prognostic value in MDSs in two studies [11,12]. In other chronic myeloid diseases, such as Philadelphia chromosome-negative myeloproliferative neoplasms, a higher AMC is associated with adverse disease features and inferior prognosis [13,14]. In MDSs, data on the prognostic impact of a subtle monocytosis below the threshold of 1.0 × 10^9^/L, defining chronic-myelomonocytic leukemia (CMML) according to the WHO 2016 classification, are scarce.

We, therefore, intended to examine the distribution of the AMC in the different MDS sub- and risk groups according to WHO 2016 and IPSS-R and to analyse its impact on overall and leukemia-free survival in a cohort of primary MDS patients, both as a continuous variable and according to different cut-offs.

## 2. Material and Methods

We screened the Düsseldorf MDS registry for patients with information on the AMC and the IPSS-R category at diagnosis. The Düsseldorf MDS registry covers virtually all newly diagnosed MDS cases in the Greater Düsseldorf area since 1982. The structure of the registry and the diagnostic criteria for inclusion of cases have been described earlier in detail [15]. Monocyte counts were obtained from manual differentials and promonocytes and monoblasts were considered as blast equivalents.

Patients with an AMC > 1.0 × 10^9^/L or a white blood cell count >13.0 × 10^9^/L were excluded, as they were likely to suffer from CMML or myelodysplastic/myeloproliferative neoplasms.

For comparative analyses, patients were stratified in three groups according to the rounded quartiles of the AMC (first and second quartile: AMC < 0.2 × 10^9^/L, third quartile: AMC 0.2–0.4 × 10^9^/L, and fourth quartile: AMC > 0.4 × 10^9^/L).

Categorical variables were analysed by frequency tables and compared using the χ2-test, continuous variables were described by median (range) and compared between different groups using the Mann–Whitney (comparison of two groups) or the Kruskal–Wallis (comparison of ≥3 groups) test. Overall and leukemia-free survival were calculated in months from the date of diagnosis to the respective event date. Time-to-event curves were calculated by the Kaplan–Meier method and the log-rank test was used for univariate comparison. Cox proportional hazard regression models were applied for multivariate analysis. A *p*-value < 0.05 was considered statistically significant.

To assess the effect of AMC on survival as a continuous variable, p-splines were estimated, both in a univariate as well as in multivariate settings. Four degrees of freedom were chosen. However, sensitivity analyses showed that this choice had only little influence on the results. Progression to AML was analysed via the competing risks methodology, with “death without progression” being the competing event. The Gray test based on the model of Fine and Gray was used to compare variables with regard to this endpoint. All analyses were performed with IBM SPSS Statistics, version 25 or R, version 4.2.1.

## 3. Results

### 3.1. Study Population

We identified a cohort of 993 patients (male *n* = 576 (58%), female *n* = 417 (42%)) with a median age of 66 years (IQR 58–73). A total of 112 cases (11.3%) were known to be therapy-related following cytotoxic treatment or radiation and seven (7%) cases represented secondary MDS evolving from another haematological neoplasm.

During follow-up (median 29 months, IQR 10–61), 633 patients (63.7%) died and 247 patients (24.9%) experienced a leukemic transformation.

Stem cell transplantation was performed in 132/993 patients (13.3%; allogeneic *n* = 129, autologous *n* = 3) and 77/993 (7.8%) patients received at least one cycle of an intensive induction chemotherapy regimen. Hypomethylating agents (HMA) were used in 85/993 patients (8.6%) and immune modulatory imide drugs (IMID) in 69/993 (6.9%). The remaining patients (630/993; 63.4%) received low-intensity treatments such as erythropoiesis stimulating agents or immunosuppression (cyclosporine ± anti-thymocyte globulin) or best supportive care, including transfusion and iron chelation. Details regarding MDS subtypes according to the WHO 2016 classification, cytogenetic risk groups according to IPSS-R and peripheral blood values, blast counts in bone marrow and peripheral blood, and additional parameters (LDH, presence or absence of bone marrow fibrosis, transfusion dependency) at the time point of diagnosis are shown in Table 1.

### 3.2. Absolute Monocyte Counts in the Study Population, Influence of Sex and Age

The median AMC of the whole population was 0.19 × 10^9^/L (IQR 0.07–0.374). It was higher in male patients compared to female patients (median 0.20 × 10^9^/L (IQR 0.09–0.42) versus 0.16 × 10^9^/L (IQR 0.07–0.32), *p* = 0.002), but not significantly affected by age </≥ 65 years (median 0.18 × 10^9^/L (IQR 0.07–0.36) versus 0.19 × 10^9^/L (IQR 0.08–0.39), *p* = 0.159).

### 3.3. Monocyte Counts in Different MDS Subgroups

In MDSs with excess blasts (*n* = 356) the AMC was significantly lower compared to MDSs without excess blasts (*n* = 637; median 0.12 vs. 0.21, *p* < 0.001). For MDSs with uni- or multi-lineage dysplasia (*n* = 511), a ring-sideroblastic phenotype (*n* = 138) was associated with a higher AMC (median 0.27 vs. 0.19 × 10^9^/L, *p* = 0.002). MDSs with del(5q) did not differ with regard to the AMC compared to MDSs non-del(5q) (median 0.18 vs. 0.21 × 10^9^/L, *p* = 0.910). There was no significant difference between primary and therapy-related MDSs (median AMC 0.19 vs. 0.19 × 10^9^/L, *p* = 0.608) and the presence or absence of fibrosis (median 0.20 vs. 0.17 × 10^9^/L, *p* = 0.351) or transfusion dependency (median 0.16 vs. 0.20 × 10^9^/L, *p* = 0.05). Details are shown in Table 2.

Following stratification according to IPSS-R, a continuous decline in the AMC with rising IPSS-R categories was noted from a median of 0.30 × 10^9^/L in very low-risk patients to a median of 0.12 × 10^9^/L in very high-risk patients, with the differences between the single IPSS-R groups being highly significant (see Figure 1 and Table 2).

### 3.4. Association of the Absolute Monocyte Count with Peripheral Blood Values and Other Disease Characteristics

An AMC < 0.2 × 10^9^/L was associated with a significantly lower haemoglobin concentration and lower numbers of neutrophils, lymphocytes, and platelets and a higher count of bone marrow blasts. In patients with a monocyte count >0.4 × 10^9^/L, higher numbers of neutrophils and lymphocytes were noted, but there was no difference with regard to haemoglobin concentrations and counts of neutrophils, platelets, and bone marrow blasts (for details, see Table 1).

### 3.5. Impact of Monocyte Count on Overall Survival

#### 3.5.1. Monocyte Count Divided According to Quartiles of the Population

Within the whole population (*n* = 993), both an AMC < 0.2 × 10^9^/L (representing the first and second quartile of the population) and >0.4 × 10^9^/L (representing the fourth quartile) were associated with a significantly shorter median overall survival compared to an AMC between 0.2 and 0.4 × 10^9^/L (median 34 or 37 versus 70 months, *p*< 0.001). However, for patients suffering from therapy-related MDSs, OS was not influenced by the AMC (median 22 or 18 versus 42 months, *p* = 0.312). The same was true for MDS patients having received HSCT or induction chemotherapy as part of their treatment (median OS 32 or 40 versus 40 months, *p* = 0.759). Therefore, further survival analyses were restricted to patients with primary MDSs, who did not receive intensive chemotherapy or stem cell transplantation (*n* = 689).

For this subgroup, an AMC < 0.2 × 10^9^/L or >0.4 × 10^9^/L was associated with a significantly shorter median overall survival compared to an AMC between 0.2 and 0.4 × 10^9^/L (median 39 or 40 versus 77 months, *p* < 0.001), with the survival curves of patients with an AMC < 0.2 × 10^9^/L and an AMC > 0.4 × 10^9^/L being superimposable, as shown in Figure 2A.

In a univariate Cox regression model, an AMC < 0.2 × 10^9^/L or > 0.4 × 10^9^/L was associated with a higher risk of death compared to an AMC between 0.2–0.4 × 10^9^/L (HR 1.61; 95% CI 1.28–2.02, *p* < 0.001; see Table 3, row 1).

After stratification according to IPSS-R, the OS according to these AMC subgroups differed significantly for intermediate-risk patients (*n* = 179; median OS 42 or 31 vs. 82 months, *p* = 0.025, see Figure 2B). Similar differences were noted for low-risk patients (*n* = 258; median OS 64 or 54 vs. 107 months) and high-risk patients (*n* = 119; median OS 25 and 18 months vs. 41 months). However, these differences were not statistically significant (low risk: *p* = 0.103; high risk *p* = 0.413). For very low-risk and very high-risk patients, no difference was observed, but patient numbers were low (*n* = 49 and *n* = 93, respectively).

In a multivariate Cox regression model together with the cytogenetic risk groups according to IPSS-R, the bone marrow blast count (< or ≥5%) and levels of cytopenias (haemoglobin < or ≥10 g/dL, absolute neutrophil count < or ≥0.8  ×  10^9^/L, and platelets < or ≥100  ×  10^9^/L), the AMC remained an independent prognostic factor. Finally, the AMC retained its independent prognostic value in a multivariate model together with the single IPSS-R risk groups and age </≥ 65 years. For details, see Table 3.

#### 3.5.2. Monocyte Count as a Continuous Variable

In a univariate proportional hazards model the effect of the AMC was analysed via p-splines. Here, we found a U-shaped effect with the lowest hazard around 0.3 × 10^9^/L (see Figure 3A). The prognostic impact of the AMC remained largely unchanged in multivariate proportional hazards models either together with the single factors of the IPSS-R (see Figure 3B and Table 4, row 2) or together with the single IPSS-R risk groups (see Figure 3C and Table 4, row 3).

#### 3.5.3. Absolute Monocyte Count and Risk of Transformation to AML

Compared to patients with an AMC between 0.2 and 0.4 × 10^9^/L, only monocytopenic patients (AMC < 0.2 × 10^9^/L) had a significantly higher risk of transformation to acute myeloid leukemia (AML) in a Fine and Gray model (HR 2.0; 95% CI 1.219–3.28, *p* = 0.006). An AMC > 0.4 × 10^9^/L was not associated with a significantly higher risk of progression to AML (HR 1.38; 95% CI 0.766–2.48, *p* = 0.280). Cumulative incidence curves depicting the probability of progression to AML with “death without AML” as competing risk are shown in Figure 4.

## 4. Discussion

Our analysis of a cohort of 993 uniformly characterized patients reveals a high prevalence of monocytopenia in MDSs. In patients with primary MDSs, who did not receive allogeneic stem cell transplantation or intensive induction chemotherapy (*n* = 689), the AMC affects prognosis independently of the IPPS-R.

Whereas in an early study, a negative impact of monocytosis was reported in MDSs [16], the current data indicate clearly a negative impact of monocytopenia [11,12]. This discrepancy can be explained by the fact that in the early study [16], which was published before the introduction of the FAB classification, 11/37 patients had an AMC > 1.5 × 10^9^/L. Those cases would now be classified as chronic myelomonocytic leukemia, making a direct comparison of the results impossible. Currently, the largest body of evidence regarding the prognostic impact of monocytopenia comes from an analysis of the Greek MDS registry [12]. In this analysis of 1719 patients, monocytopenia, defined as an AMC < 0.2 × 10^9^/L, was associated with an inferior leukemia-free survival and was described as an adverse risk factor independently from the IPSS-R. Our results from the Düsseldorf MDS registry are fully in line with these observations. A large monocentric study in the USA (*n* = 889) found a negative prognostic impact of monocytopenia, however, only in a univariate analysis and not within the context of the IPSS-R. This may be explained by a higher cut-off used to define monocytopenia (AMC < 0.3 × 10^9^/L).

According to our results, monocytopoiesis seems to be impaired in a considerable proportion of MDS patients. With a median of 0.198 × 10^9^/L, the AMC in our cohort is considerably lower than the AMCs that have been reported for healthy Caucasian adult men (0.34 × 10^9^/L) and women (0.30 × 10^9^/L) [17]. There is no general consensus about the definition of monocytopenia, but with regard to a commonly used cut-off of 0.2 × 10^9^/L [18] nearly 50 percent of our cohort has to be considered monocytopenic.

Currently, the degree of cytopenias is assessed using binary cut-offs, which are by nature always somewhat arbitrary. Some information may, therefore, be lost. Hence, in the newest version of the IPSS (IPSS-M), cytopenias are for the first time taken into account as continuous variables. If analysed as a continuous variable, the AMC showed a U-shaped curve with the lowest hazard around 0.3 × 10^9^/L, which corresponds to the median AMC reported for healthy Caucasians [17].

We observed significant differences with regard to the AMC between different MDS subtypes, for example, a higher AMC in MDSs without excess blasts compared to MDSs with excess blasts and in MDSs with ringed sideroblasts (MDS-RS-SLD/MLC) compared to MDSs without ringed sideroblasts (MDS-SLD/MLD). In addition, there was a significant difference regarding the AMC between the IPSS-R subcategories, with the AMC falling continuously with rising IPSS categories. Furthermore, MDS patients with an AMC < 0.2 × 10^9^/L had significantly more severe cytopenias as well. Finally, yet importantly, monocytopenia was associated with a considerably higher risk of transition to acute myeloid leukemia. These observations hint at a direct relationship between MDS pathophysiology and monocytopenia in MDSs.

Systematic studies on the clonal involvement of the monocytic compartment in MDS patients are scarce and mainly report the results of restriction length fragment analyses of X-linked genes. In the small cohorts analysed, monocytes have been consistently shown to be clonal [19,20,21]. In addition, in MDS cases with monosomy 7, the respective cytogenetic abnormality has been described in both granulocytes and monocytes [22]. It is, therefore, likely that monocytopenia in MDS is due to clonal and ineffective monocytopoiesis. However, given the many changes in bone marrow microenvironment described in MDSs [23], a non-clonal suppression of monocytopoiesis may contribute to monocytopenia as well [24]. Even if monocytopenia in MDSs somehow reflects ineffective haematopoiesis, it represents an additional risk factor independent of the “classical” cytopenias (anemia, thrombocytopenia, and neutropenia), which are routinely taken into account by the IPSS-based scoring systems.

Monocytopenia is a hallmark of some rare inherited myeloid disorders with GATA2-deficiency, such as the MonoMAC (monocytopenia and mycobacterium avium complex infections) syndrome and the dendritic cell, monocyte, and lymphocyte deficiency, DCML [25]. Somatic GATA-2 mutations occur in sporadic forms of myeloid neoplasms as well and have been shown to be associated with monocytopenia, especially if the C-terminal zinc-finger domain is involved [26]. However, the frequency of GATA2 mutations in MDSs is low, with a frequency in the range of 1–2% [27,28]. Hence, GATA2 mutations cannot explain the high prevalence of monocytopenia in MDS patients. Whether monocytopenic MDS patients differ from non-monocytopenic MDS patients with regard to their mutational profile is currently unknown.

As phagocytes do, monocytes and neutrophils share some overlapping functions in host defence, especially against bacterial and fungal infections [29]. Recent reports imply that monocytes from MDS patients exhibit relatively normal innate immune functions compared to monocytes from healthy controls, at least when LPS-induced cytokine production and phagocytosis are assessed in vitro [30,31]. In low- and intermediate-risk MDS patients, an increased TNF-α production upon stimulation with LPS has been reported as well, together with an increased proportion of intermediate monocytes (CD14^bright^/CD16+) and a lower proportion of classical monocytes (CD14^bright^/CD16−) [32]. It is, therefore, likely that monocytes can in part substitute for reduced or defective neutrophils. Hence, monocytopenia in MDS probably goes along with the loss of an additional line of antimicrobial defence and confers an increased risk of infections. However, reduced effector functions of monocytes [33] and monocyte-derived macrophages [34] have been reported as well in MDSs, indicating that more research is needed to clarify the functional capacity of cells of the mononuclear phagocyte system in MDSs, especially since existing data point towards a considerable heterogeneity between certain MDS subtypes [30,33].

Surprisingly, our analysis showed that not only is monocytopenia associated with an increased mortality, but a subtle monocytosis (>0.4 × 10^9^/L) as well. The majority of the respective MDS cases would now be classified as CMML according to both the recently revised WHO classification [4] and the newly proposed “international consensus classification” (ICC) [5], since “oligomonocytic” CMML with an AMC > 0.5 × 10^9^/L and at least 10% monocytes in the peripheral blood with a clonal marker and “classical” CMML share largely overlapping features [35,36,37]. Mutations in genes involved in RNA splicing and epigenetic modification (e.g., *SRSF2*, *U2AF1*, *TET2*, and/or *ASXL1*) are a frequent event in CMML [38]. According to IPSS-M, mutations in *SRSF2*, *U2AF1*, and *ASXL1* are associated with an adverse prognosis in MDSs [9]. The negative prognostic impact of a subtle monocytosis in MDSs could, therefore, at least partially be due to an accumulation of adverse mutations in this patient group. However, an elevated risk of AML transformation was associated only with monocytopenia, but not with subtle monocytosis.

Cells of the monocyte/macrophage system play a crucial role in atherosclerosis and heart failure [39,40]. A higher AMC is associated with an increased cardiovascular risk both in the general population [41,42] and several medical conditions, such as end stage kidney disease [43] and chronic HIV infection [44], as shown recently. Therefore, a higher monocyte count could be an indicator of an increased cardiovascular risk in MDSs. Within this context, observations regarding clonal haematopoesis of undetermined potential (CHIP) are of importance, which demonstrate an increased expression of inflammatory genes in innate immune cells, such as monocytes and macrophages, in this condition, which provides a potential explanation for the epidemiological link between mutations present in myeloid stem cells and an increased cardiovascular risk [45]. However, further studies are necessary to prove this hypothesis and to assess other potential sequelae of a raised AMC in MDSs.

From a clinical point of view, considering the AMC in MDS care seems to be of value in two ways. For transplant-eligible patients, especially in the intermediate-risk group according to IPSS-R, a low AMC could provide an additional argument for allogeneic stem cell transplantation, especially in combination with other risk factors such as elevated LDH [46], bone marrow fibrosis [47], or transfusion dependency [48]. Given its association with AML transformation, a low AMC could support the use of hypomethylating agents as well, as already proposed [12]. Especially for lower-risk MDS patients with a higher AMC, proper attention should be paid to modifiable cardiovascular risk factors.

### Limitations

One major drawback of our study is the lack of molecular data, which prevents a study of the association of the mutational profile with the AMC, and assessing whether the AMC still provides prognostic information when the molecular risk profile is already considered. Furthermore, given the long history of the Düsseldorf MDS registry, the vast majority of non-transplant-eligible patients were treated with best supportive care, and only a minority received modern agents such as IMIDs or HMA. This fact prevents the assessment of a potential impact of these common treatments on the prognostic role of the AMC.

## 5. Conclusions

Our data clearly demonstrate a prognostic impact of the AMC in MDSs, which is independent from the IPSS-R. Surprisingly, we could demonstrate that both monocytopenia and a subtle monocytosis affect overall survival. Whereas monocytopenia was clearly associated with established adverse prognostic features and risk of AML progression, this was not the case for a subtle monocytosis. Hence, the factors mediating the negative prognostic impact of a higher AMC in MDSs remain to be elucidated.

## Figures and Tables

**Figure 1 cancers-15-03572-f001:**
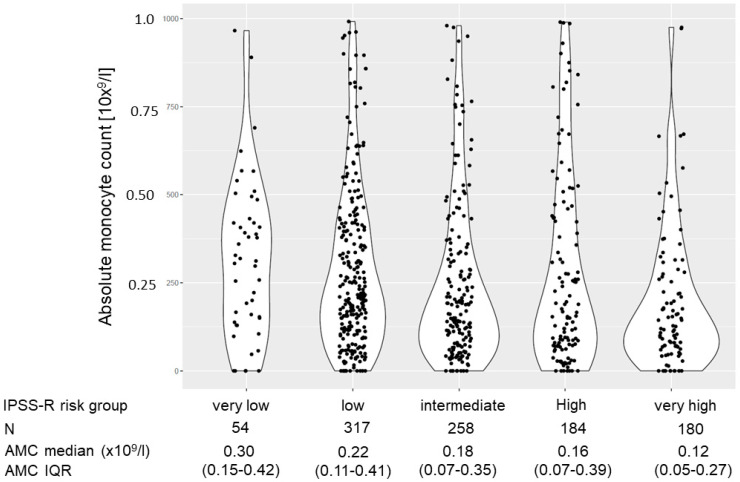
Absolute monocyte count in the single risk groups according to IPSS-R.

**Figure 2 cancers-15-03572-f002:**
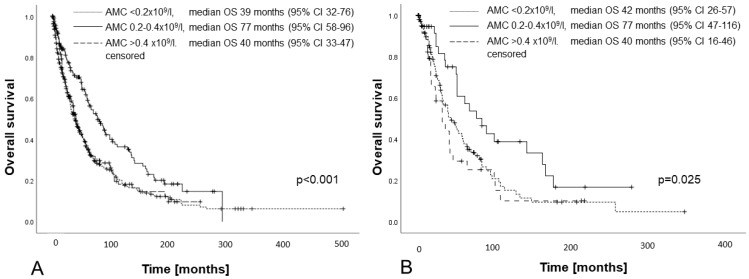
Survival of patients with primary MDSs, without induction chemotherapy or stem cell transplantation, stratified by the absolute monocyte count (<0.2 × 10^9^/L, 0.2–0.4 × 10^9^/L, and >0.4 × 10^9^/L). (**A**) Whole cohort (*n* = 689); (**B**) IPSS-R intermediate risk (*n* = 258).

**Figure 3 cancers-15-03572-f003:**
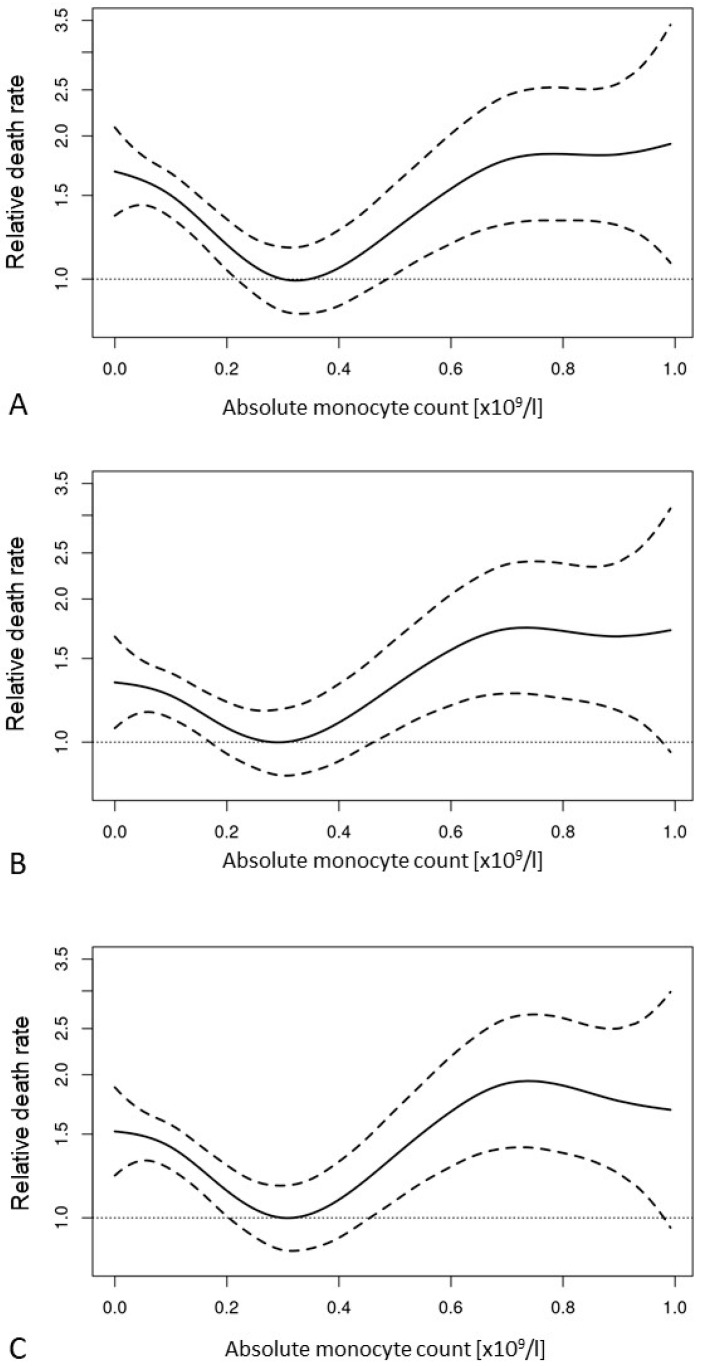
P-splines modelling the prognostic impact of the absolute monocyte count in a univariate proportional hazards model (**A**) and two multivariate proportional hazards models together with the single factors of the IPSS-R (**B**), see also Table 4, row 2 or the single IPSS-R risk groups (**C**), see also Table 4, row 3.

**Figure 4 cancers-15-03572-f004:**
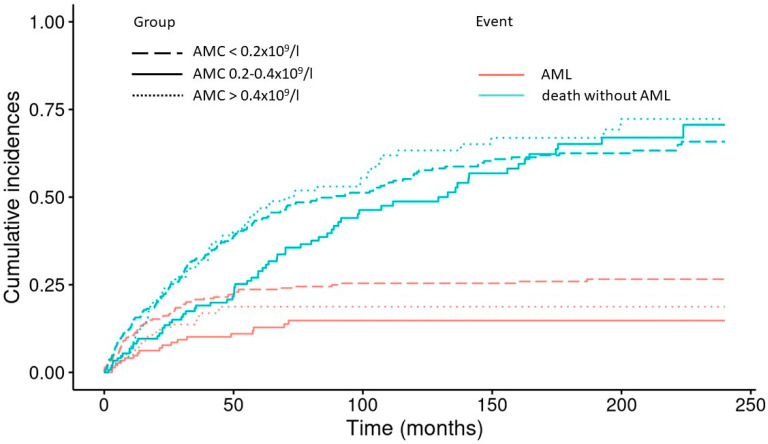
Cumulative incidence curves depicting the probability of progression into AML of patients with primary MDSs, without induction chemotherapy or stem cell transplantation, stratified by the absolute monocyte count (<0.2 × 10^9^/L (*n* = 529), 0.2–0.4 × 10^9^/L (*n* = 241), and >0.4 × 10^9^/L (*n* = 223)) and “death without progression” as the competing event.

**Table 1 cancers-15-03572-t001:** Patient characteristics in the whole cohort and according to the absolute monocyte count (group A < 0.2 × 10^9^/L, group B 0.2–0.4 × 10^9^/L, group C > 0.4 × 10^9^/L).

		Group A	Group B	Group C	*p*	
	All	AMC <0.2 × 10^9^/L	AMC 0.2–0.4 × 10^9^/L	AMC > 0.4 × 10^9^/L	A vs. B	B vs. C
WHO 2016, *n* (%)	993	529	241	223		
MDS-SLD	71 (7.2)	38 (7.2)	18 (7.5)	15 (6.7)		
MDS-MLD	302 (30.4)	157 (29.7)	70 (29)	75 (33.6)		
MDS-RS-SLD	46 (4.6)	15 (2.8)	17 (7.1)	14 (6.3)		
MDS-RS-MLD	92 (9.3)	37 (7)	24 (10)	31 (13.9)		
MDS(del5q)	115 (11.6)	54 (10.2)	46 (19.1)	15 (6.7)		
MDS-EB-1	162 (16.3)	86 (16.3)	34 (14.1)	42 (18.8)		
MDS-EB-2	194 (19.5)	138 (26.1	29 (12)	27 (12.1)		
MDS-U	11 (1.1)	4 (0.8)	3 (1.2)	4 (1.8)		
IPSS-R, *n* (%)						
Very low	54 (5.4)	19 (3.6)	18 (7.5)	17 (7.6)		
Low	317 (31.9)	141 (26.7)	93 (38.6)	83 (37.2)		
Intermediate	258 (26.0)	146 (27.6)	62 (25.7)	50 (22.4)		
High	184 (18.5)	104 (19.7)	35 (14.5)	45 (20.2)		
Very high	180 (18.1)	119 (22.5)	33 (13.7)	28 (12.6)		
Cytogenetic risk groups (IPSS-R)						
Very good	34 (3.4)	14 (2.6)	5 (2.1)	15 (6.7)		
Good	609 (61.3)	321 (60.7)	153 (63.5)	135 (60.5)		
Intermediate	165 (16.6)	95 (18)	42 (17.4)	28 (12.6)		
Poor	72 (7.3)	33 (6.2)	17 (7.1)	22 (9.9)		
Very poor	113 (11.4)	66 (12.5)	24 (10)	23 (10.3)		
Demographics						
Age, median (years), IQR	66 (58–73)	66 (58–73)	66 (58–73)	67 (59–73)	0.782	0.220
Male, *n* (%)	576 (58)	285 (53.9)	138 (57.3)	153 (68.6)	0.381	0.012
Leukemic transformation, *n* (%)	247 (24.9)	145 (27.4)	55 (22.8)	47 (21.1)		
Deaths, *n* (%)	633 (63.7)	359 (67.9)	134 (55.6)	140 (62.8)		
Lost to follow-up, *n* (%)	35 (3.5)	20 (3.8)	9 (3.7)	6 (2.7)		
Peripheral blood values						
Haemoglobin, median (g/L), IQR	95 (83–110)	93 (82–109)	99 (84–113)	98 (85–112)	0.018	0.79
Neutrophil count, median (×10^9^/L), IQR	1.59 (0.82–2.66)	1.12 (0.63–2.05)	1.84 (1.20–2.66)	2.54 (1.48–3.75)	<0.001	<0.001
Lymphocyte count, median (×10^9^/L), IQR	1.21 (0.79–1.68)	1.1 (0.70–1.54)	1.26 (0.83–1.73)	1.47 (0.99–2.05)	0.006	<0.001
Platelet count, median (×10^9^/L), IQR	115 (58–232)	97 (53–190)	151 (74–266)	128 (65–263)	<0.001	0.227
LDH						
Data available, *n* (%)	851 (85.7)	461 (87.1)	204 (84.6)	186 (83.4)		
LDH, median (U/L)	199 (172–256)	203 (170–257)	198 (177–243)	198 (172–268)	0.798	0.917
LDH > ULN (240 U/L), *n* (%)	259 (26.1)	149 (28.2)	51 (16.2)	59 (31.7)	0.058	0.141
Blasts						
Blasts bone marrow, median (%), IQR	3 (1–8)	4 (2–10)	2 (1–5)	3 (1–7)	<0.001	0.394
Blasts peripheral blood, median (%), IQR Range	0 (0–0) (0–19)	0 (0–0) (0–19)	0 (0–0) (0–18)	0 (0–0) (0–19)	0.484	0.382
Bone marrow fibrosis						
Data available, *n* (%)	539 (54.3)	299 (56.5)	117 (48.5)	123(55.2)		
With fibrosis, *n* (%)	75 (12.6)	39 (13.0)	19 (16.2)	17 (13.8)	0.398	0.600
Transfusion dependent						
Data available, *n* (%)	415 (41.8)	229 (43.7)	94 (39)	92 (41.3)		
Transfusion dependent, *n* (%)	218 (51.8)	129 (56.3)	45 (47.9)	44 (47.8)	0.166	0.995

MDS: myelodysplastic syndrome; SLD: single lineage dysplasia; MLD: multi-lineage dysplasia; RS: ringed sideroblasts; EB: excess blasts; MDS-U: myelodysplastic syndrome unclassifiable; IQR: inter-quartile range, LDH: lactate dehydroxygenase.

**Table 2 cancers-15-03572-t002:** Absolute monocyte count in different MDS subgroups.

	*n*	AMC (×10^9^) Median (IQR)	*p*
Total	993	0.19 (0.07–0.37)	
WHO 2016			
MDS-SLD	71	0.19 (0.11–0.33)	<0.001
MDS-MLD	302	0.19 (0.09–0.40)	
MDS-RS-SLD	46	0.31 (0.13–0.52)	
MDS-RS-MLD	92	0.26 (0.15–0.50)	
MDS(del5q)	115	0.21 (0.10–0.32)	
MDS-EB-1	162	0.16 (0.06–0.45)	
MDS-EB-2	194	0.10 (0.04–0.26)	
MDS-U	11	0.32 (0.09–0.59)	
IPSS-R			
Very low	54	0.30 (0.15–0.42)	<0.001
Low	317	0.22 (0.11–0.41)	
Intermediate	258	0.18 (0.07–0.35)	
High	184	0.16 (0.07–0.39)	
Very high	180	0.12 (0.05–0.27)	
MDS with excess blasts	356	0.12 (0.05–0.32)	<0.001
MDS without excess blasts	637	0.21 (0.11–0.40)	
MDS del(5q)	115	0.21 (0.09–0.32)	0.910
MDS non-del(q)	878	0.18 (0.07–0.39)	
MDS-SLD/MLD	373	0.19 (0.10–0.39)	0.002
MDS-RS-SLD/MLD	138	0.27 (0.13–0.50)	
Lower-risk MDSs (IPSS-R very low/low)	371	0.24 (0.11–042)	<0.001
Higher-risk MDSs (IPSS-R intermediate/high/very high)	622	0.15 (0.06–0.33)	
Therapy-related MDSs	112	0.19 (0.09–0.39)	0.608
Primary MDSs	881	0.19 (0.07–0.37)	
Transfusion dependent	218	0.16 (0.06–0.33)	0.05
Transfusion independent	197	0.20 (0.09–0.40)	
Without bone marrow fibrosis	464	0.17 (0.07–0.38)	0.351
With bone marrow fibrosis	75	0.20 (0.07.0.38)	

IQR: inter-quartile range; MDS: myelodysplastic syndrome; SLD: single lineage dysplasia; MLD: multi-lineage dysplasia; RS: ringed sieroblasts; EB: excess blasts; MDS-U: myelodysplastic syndrome unclassifiable.

**Table 3 cancers-15-03572-t003:** Factors affecting prognosis in MDS patients in univariate analysis with the absolute monocyte count (AMC) as a dichotomous variable (<0.2 × 10^9^/L or >0.4 × 10^9^/L versus 0.2–0.4 × 10^9^/L) and three multivariate models (multivariate I: AMC together with the degree of cytopenias, bone marrow blast count, and cytogenetic risk groups according to IPSS-R; multivariate II: AMC together with the single IPSS-R risk groups; multivariate III: AMC together with the single IPSS-R risk groups, age > 65 and male sex).

	Univariate			Multivariate I			Multivariate II			Multivariate III		
	HR	95% CI	*p*	HR	95% CI	*p*						
Age > 65	2.37	1.92; 2.94	<0.001							2.45	1.97; 3.04	<0.001
Male sex	1.31	1.08; 1.58	0.006							1.10	0.91; 1.34	0.332
Hb < 100 g/L	1.48	1.23; 1.79	<0.001	1.59	1.31; 1.94	<0.001						
Neutrophils < 0.8 × 10^9^/L	1.74	1.32; 2.30	<0.001	0.92	0.69; 1.24	0.582						
Platelets < 100 × 10^9^/L	1.56	1.23; 1.99	<0.001	1.59	1.30; 1.94	<0.001						
Bone marrow blasts >5%	2.81	2.29; 3.45	<0.001	2.16	1.72.; 2.70	<0.001						
IPSS-R cytogenetic risk category												
good vs. very good	1.67	0.86; 3.25	0.130	2.13	1.09; 4.15	0.027						
intermediate vs. very good	2.48	1.24; 4.96	0.010	2.80	1.40; 5.61	0.004						
poor vs. very good	2.55	1.22; 5.32	0.014	2.61	1.25; 5.48	0.011						
very poor vs. very good	8.15	4.01; 16.55	<0.01	7.18	3.49; 14.78	<0.001						
IPSS-R category												
low vs. very low	1.44	0.92; 2.25	0.113				1.43	0.91;2.24	0.119	1.57	1.00; 2.46	0.050
intermediate vs. very low	2.16	1.37; 3.41	<0.001				2.13	1.35; 3.36	0.001	2.22	1.40; 3.50	0.001
high vs. very low	4.03	2.51; 6.46	<0.001				3.90	2.43; 6.26	<0.001	3.85	2.34; 6.19	<0.001
very high vs. very low	7.10	4.41; 11.43	<0.001				6.81	4.23; 10.97	<0.001	7.35	4.56; 11.85	<0.001
Monocyte count <0.2 × 10^9^/L or >0.4 × 10^9^/L	1.61	1.28–2.02	<0.001	1.32	1.04–1.67	0.021	1.47	1.16; 1.85	0.001	1.63	1.29; 2.05	<0.001

**Table 4 cancers-15-03572-t004:** Factors affecting prognosis in MDS patients in univariate analysis with the absolute monocyte count as a continuous variable (see Figure 3A–C) and two multivariate models (multivariate I: AMC together with the degree of cytopenias, bone marrow blast count, and cytogenetic risk groups according to IPSS-R; multivariate II: AMC together with the single IPSS-R risk groups).

	Univariate			Multivariate I			Multivariate II		
	HR	95% CI	*p*	HR	95% CI	*p*	HR	95% CI	*p*
Hb < 10 g/dL	1.48	1.23; 1.79	<0.001	1.59	1.30; 1.94	<0.001			
Neutrophil count < 0.5 × 10^9^/L	1.74	1.32; 2.30	<0.001	0.94	0.70; 1.26	0.684			
Platelet count < 100 × 10^9^/L	1.56	1.23; 1.99	<0.001	1.67	1.36; 2.05	<0.001			
Bone marrow blasts > 5%	2.81	2.29; 3.45	<0.001	2.09	1.66; 2.63	<0.001			
IPSS-R cytogenetic risk category									
good vs. very good	1.67	0.86; 3.25	0.130	2.13	1.09; 4.17	0.027			
intermediate vs. very good	2.48	1.24; 4.96	0.010	2.81	1.40; 5.66	<0.001			
poor vs. very good	2.55	1.22; 5.32	0.014	2.68	1.28; 5.63	<0.001			
very poor vs. very good	8.15	4.01; 16.55	<0.01	7.45	3.61; 15.37	<0.001			
IPSS-R category									
low vs. very low	1.44	0.92; 2.25	0.113				1.43	0.91; 2.25	0.119
intermediate vs. very low	2.16	1.37; 3.41	<0.001				2.14	1.35; 3.39	0.001
high vs. very low	4.03	2.51; 6.46	<0.001				3.82	2.37; 6.14	<0.001
very high vs. very low	7.10	4.41; 11.43	<0.001				7.23	4.48; 11.67	<0.001
Absolute monocyte count as a continous variable	See Figure 3A			See Figure 3B			See Figure 3C		

## Data Availability

The datasets used and/or analysed during the current study available from the corresponding author on reasonable request.

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
