# Peer review of "The Absolute Monocyte Count at Diagnosis Affects Prognosis in Myelodysplastic Syndromes Independently of the IPSS-R Risk Score"

_cancers, 2023, doi:10.3390/cancers15143572_

Round 1

Reviewer 1 Report

Your manuscript is about the impact of absolute monocyte count in patients with Myelodysplastic Syndrome. Data were provided by the Düsseldorf MDS-registry (n=993). Patients were grouped in absolute monocyte count at diagnosis <0.2x109/l, 0.2-0.4x109/l and >0.4x109/l. Furthermore, absolute monocyte count as a continuous variable was used. Your manuscript shows a negative impact of low absolute monocyte count as well as monocyte count >0.4x109/l at time of diagnosis. Even more, your group found monocyte count as a separate prognostic factor independent from IPSS-R. Your work is of enormous interest for the scientific field helping to better understand risk stratification in patients with Myelodysplastic Syndrome. Following you will find some major and minor issues regarding your manuscript.

Major Issue:

1.     How was monocyte count measured? Automatically while measuring CBC counts or manually? Was the same method used for all patients?

2.     Regarding Table 1: No patients of your cohort seem to have peripheral blasts. Are peripheral blasts indicated for every patient?

3.     Line 57: Please further discuss if monocytopenia is “only” representing (myeloid) cytopenia in general.

4.     Line 78: Please discuss if peripheral blasts can be counted as monocytes by using your method although it may not be a major issue of your study (line 138-139 (results) and line 88-89 (discussion).

5.     Line 90-93: Do you have information regarding relevant comorbidities?

Minor Issues:

1.     Line 31 and line 99 (discussion): Remove the space.

2.     Line 70: Please adapt the text formatting.

3.     Table 1 and Table 2: Please explain the mentioned abbreviations. 

4.     Table 1: Please correct “IPSS-R Cytogenetic risk groups”

5.     Figure 4: Please discuss if you want to mention the group size “n=” in the figure legend.

Good quality

Reviewer 2 Report

The paper entitled “The Absolute Monocyte Count at Diagnosis Affects Prognosis 2 in Myelodysplastic Syndromes Independently of the IPSS-R Risk Score” have its strength showing correlation data about monocyte count and prognostic in MDS patients, which is a controversial theme in the literature, as some papers observed association with low and other with high monocyte counts.

In the introduction, some information about genetic mutations as prognostic factors in MDS (TET2, SRSF2, ASXL1, DNMT3A) will add significant value to the work, especially because the theme is mentioned in the discussion. We will also encourage the authors to cite how many types of MDS exist in the introduction, as in the results they will show this stratification. 

The authors should discuss more about the uncertainty about prognostic factors associated with monocyte counts as the literature on this data is controversial. In the introduction, the authors cited two works associating low monocyte counts with predictive value in MDS. However, some papers saw the association of high monocyte counts with poor prognostic (10.1159/000207157; 10.1155/2016/5479013), and others with low monocyte counts and an increased risk to secondary AML from MDS (10.1038/bcj.2017.30). Another thing that is missing in the discussion is the analysis of functional aspects of monocytes to see how their response are (10.1016/j.cellimm.2016.07.005; 10.1002/JLB.5AB1017-419RR; 10.1016/j.bcmd.2020.102507; 10.1155/2016/5479013), as low monocytes counts could have functional or reactive cells that will answer as well as normal monocytes.

Minor revision is required.

Reviewer 3 Report

This paper is weel done, with an interesting issue. The  prognostic impact of AMC in myelodysplastic syndromes  is less well studied and the  potential prognostic value, expecially in the setting of patients candiate to allotransplant is very important for clinicians and could be able to help them in their daily activities. 

An important limitations of this study is  the lack of molecular data. as suggested by the authors, also. The authors, in a subsequent work could focused on this issue. 
